# The Effect of Prior Creatine Intake for 28 Days on Accelerated Recovery from Exercise-Induced Muscle Damage: A Double-Blind, Randomized, Placebo-Controlled Trial

**DOI:** 10.3390/nu16060896

**Published:** 2024-03-20

**Authors:** Shota Yamaguchi, Takayuki Inami, Hiroyuki Ishida, Akihisa Morito, Satoshi Yamada, Naoya Nagata, Mitsuyoshi Murayama

**Affiliations:** 1Institute of Physical Education, Keio University, Yokohama 223-8521, Japan; inamit@keio.jp (T.I.); nnagata@keio.jp (N.N.); murayama@z3.keio.jp (M.M.); 2Sports Medicine Research Center, Keio University, Yokohama 223-8521, Japan; rxb03102@nifty.ne.jp; 3Taisho Pharmaceutical Co., Ltd., Saitama 331-9530, Japan; a-morito@taisho.co.jp (A.M.); sat-yamada@taisho.co.jp (S.Y.)

**Keywords:** creatine monohydrate, exercise-induced muscle damage, eccentric exercise, dietary supplementation

## Abstract

Despite the known beneficial effects of creatine in treating exercise-induced muscle damage (EIMD), its effectiveness remains unclear. This study investigates the recovery effect of creatine monohydrate (CrM) on EIMD. Twenty healthy men (21–36 years) were subjected to stratified, randomized, double-blind assignments. The creatine (CRE) and placebo (PLA) groups ingested creatine and crystalline cellulose, respectively, for 28 days. They subsequently performed dumbbell exercises while emphasizing eccentric contraction of the elbow flexors. The EIMD was evaluated before and after exercise. The range of motion was significantly higher in the CRE group than in the PLA group 24 h (h) post exercise. A similar difference was detected in maximum voluntary contraction at 0, 48, 96, and 168 h post exercise (*p* = 0.017–0.047). The upper arm circumference was significantly lower in the CRE group than in the PLA group at 48, 72, 96, and 168 h post exercise (*p* = 0.002–0.030). Similar variation was observed in the shear modulus of the biceps brachii muscle at 96 and 168 h post exercise (*p* = 0.003–0.021) and in muscle fatigue at 0 and 168 h post exercise (*p* = 0.012–0.032). These findings demonstrate CrM-mediated accelerated recovery from EIMD, suggesting that CrM is an effective supplement for EIMD recovery.

## 1. Introduction

Frequent strenuous physical activity, including eccentric contractions that exert tension while stretching, causes exercise-induced muscle damage (EIMD) characterized by multiple physiological disruptions, including myocyte damage, impaired excitation–contraction coupling, disorganization of myofibrillar contractile mechanisms, and structural alterations in the extracellular matrix [1]. Consequently, reduced muscle strength and range of motion, increased muscle stiffness, and muscle soreness occur [2]. These symptoms typically appear within 24 h post exercise and may persist for five days or longer [3]. Magnetic resonance imaging analysis has indicated that muscle tissue swelling can persist for up to one month, necessitating extended recovery periods depending on the injury severity [4]. Furthermore, EIMD alters electromyographic patterns, including kinematic alterations and increased coordinated muscle activity, which may increase the risk of muscle strain [5]. Hence, effective recovery strategies are crucial, particularly for athletes engaged in concurrent training and competition.

Owing to their complex annual training and performance schedule, athletes often use various ergogenic aids to expedite recovery from damage and sustain their physical condition [6,7]. Creatine monohydrate (CrM) amplifies hypertrophy and maximal strength gain during strength training [8]. CrM is synthesized endogenously from amino acids such as glycine, methionine, and arginine or ingested with food or as a dietary supplement [9]. Furthermore, approximately 95% of dietary CrM is stored in skeletal muscles, whereas the remaining 5% is distributed to the brain, testes, and heart [10]. CrM accumulation in skeletal muscles can mitigate post-exercise muscle damage. The modulation of mitochondrial permeability and stabilization of myocyte membranes through enhanced intramuscular phosphocreatine content are associated with this recovery process [11]. Furthermore, CrM inhibits decreases in maximum strength (i.e., one-repetition maximum) and increases in creatine kinase (CK) levels [12].

CrM can positively impact performance. For instance, Rosene et al. [13] reported that after several days of eccentric leg extension, a creatine-supplemented group exhibited a higher maximal isometric force score than the placebo group. Moreover, CrM was found to restore decreased maximal muscle strength and joint range of motion earlier than in the placebo group. Collectively, these effects can improve quality of life and maintain athletic performance. However, these findings are not universally corroborated due to the absence of sample size validation and rigorous validation through suitable methods, such as double-blind randomized controlled trials [14].

Accordingly, this study aims to clarify whether pre-consumption of CrM reduces EIMD damage and facilitates the recovery process using a statistically robust sample size. Moreover, the intrinsic characteristics associated with EIMD warranted random and hierarchical random assignments.

## 2. Materials and Methods

### 2.1. Participants

This study was conducted in accordance with the Declaration of Helsinki (21-003), and informed written consent was obtained from all participants. However, before obtaining written informed consent, participants were briefed on the nature and purpose of the experimental procedure and the associated risks.

The sample size was calculated using the appropriate input parameters (effect size: 0.3, alpha: 0.05, power: 0.8) in a two-way analysis of variance (ANOVA; G*Power version 3.1, Heinrich Heine Universität, Düsseldorf, Germany). Data from a previous study that evaluated eccentric exercise similar to the present study indicated that ten participants were necessary for each group [15]. Thus, in the current study, 20 healthy men (21–36 years old) participated in the study, with 10 per group. Participants were limited to Japanese men because EIMD symptoms are significantly influenced by age, sex, genetics, ethnicity, and exercise experience [16]. Thirty-six participants expressed their willingness to participate; however, four withdrew for personal reasons. All enrolled individuals participated in the study between July 2021 and March 2022 at the Keio Physical Education Research Institute in Kanagawa, Japan. As EIMD is influenced by training experience and age, participants were equally randomized after prior measurements of maximal muscle strength and age determination. Randomization was conducted using Predictive Analytics Software version 28 for Windows (SPSS Japan Inc., Tokyo, Japan).

Of those who completed the experiment, 12 were excluded based on previously defined exclusion criteria [2,12], namely, if the participants (1) were <20 years old, (2) were on a weight-loss diet or medication, (3) were diagnosed with acute or chronic illness, (4) were on ongoing medical treatment that may affect the immune system, (5) were recovering from injury, (6) were self-reported smokers, or (7) if they had a history of chronic alcohol abuse. The remaining 20 participants were included in the analyses (Figure 1).

All participants who completed the study had no adverse health effects attributable to CRE or PLA ingestion. Participant characteristics in the CRE group were as follows: aged 23.5 ± 4.1 years (21–36 years) and body weight 72.1 ± 11.2 kg (53–86 kg). The PLA group characteristics were as follows: aged 23.9 ± 1.9 years (21–28 years) and body weight 70.4 ± 9.9 kg (55–93 kg). No significant differences were detected in any of the baseline variables between the PLA and CRE groups (Table 1).

### 2.2. Experimental Design

In this randomized, double-blind, placebo-controlled clinical trial, participants ingested 3 g of creatine (CLE) or microcrystalline cellulose (CLE; a tasteless and odorless substance) per day. Participants were instructed to ingest the test meal with water for 28 days. CRE and PLA were each packed in aluminum opaque sachets, making the samples indistinguishable. CRE was purchased from AlzChem Trostberg GmbH (Trostberg, Germany) after being packaged and stored at room temperature. Daily consumption was self-recorded in a diary to assess the compliance rate. Following the 28-day ingestion period, participants performed the eccentric exercises. Before the experiment, the participants attended a familiarization session that included muscle strength measurements. On the first day, a pre-exercise warm-up was conducted, and an eccentric exercise involving the elbow flexor muscles was performed. Exercise tasks were safely performed as previously described [17]. The maximum voluntary contraction (MVC), range of motion (ROM), muscle soreness, muscle fatigue, circumference, muscle shear modulus, and urinary titin fragment (UTF) were measured immediately before and after the eccentric exercise routine. Measurements were repeated at 1, 24, 48, 72, 96, and 168 h post exercise. All participants performed the exercises using their left arm.

### 2.3. Eccentric Exercise

We adopted the methodology of the exercise task reported by Nosaka et al. [18], in which the participants were seated on an arm-curl bench with the hip flexed at 85° (0° = full hip extension). They completed five sets of ten eccentric exercises with dumbbells weighing 50% of the elbow joint MVC of the left arm (recorded in the familiarization session). Eccentric exercises were performed by extending the elbow joint from 90° to 180° (180° = full extension) at a 60-beat-per-minute rhythm of the metronome (i.e., extended 90° in five seconds; Figure 2). The examiner supported the elbow flexion during the concentric phase. All actions were repeated every three seconds, and a recovery period of two minutes was provided between each set.

### 2.4. Maximum Voluntary Contraction Evaluation

Maximal isometric elbow flexion strength was evaluated during 5 s isometric maximal voluntary contractions performed at an elbow angle of 90° using a handheld dynamometer (Mobie; SAKAI Medical Co., Ltd., Tokyo, Japan). Two or three (if the difference between the two measurements exceeded 10%) trials were performed, and the maximum value obtained was considered for evaluation.

### 2.5. Active Range of Motion

The centers of the acromion, lateral epicondyle, and ulnar styloid were marked using a semi-permanent marker. The elbow joint angle was photographed in relaxed and flexed states to determine the active ROM. The angle between the line connecting the center of the acromion and the lateral epicondyle and that connecting the lateral epicondyle and ulnar styloid was measured using ImageJ software (version 1.39, U. S. National Institutes of Health, Bethesda, MD, USA). The angle at the relaxed state was subtracted from that at the flexed state to determine the ROM of the elbow joint.

### 2.6. Subjective Evaluation

Muscle soreness (SOR) and fatigue were measured as subjective evaluations of muscle damage [19]. SOR and muscle fatigue were assessed using a 100 mm visual analog scale (VAS); 0 and 100 indicated no pain (or no fatigue) and extreme pain (or extreme fatigue), respectively. The SOR was measured when the participants actively extended their arms. Participants held their shoulder joints at 90° flexion and elbow joints at 180° active extension and marked the levels of perceived soreness on the VAS. The assessment of muscle fatigue was performed per previously described protocols [2].

### 2.7. Circumference

The circumference of the upper arm at 50% and 75% of the acromial process of the scapula was measured using a measuring tape (Model R-280; Futaba, Mobara, Japan) while the arm was hung down by the side. The mean of two measurements was recorded.

### 2.8. Muscle Shear Modulus

The shear modulus (SM) of the biceps brachii muscle was measured while participants were relaxed in a supine position on a bed at a 180° elbow joint, 10° shoulder joint extension, and 30° shoulder joint abduction. The ultrasonographic apparatus used an ultrasound shear wave scanner in “shear wave” mode coupled with a linear array transducer (Aplio 300; Canon Co., Ltd., Tokyo, Japan). The ultrasound transducer was placed over the muscle belly of the long head of the biceps brachii (i.e., approximately 50% of the upper arm length ranging from the acromial process of the scapula to the lateral epicondyle of the humerus) [20]. The probe was attached to the same location across sessions and days using a semi-permanent ink marker; measurement marks were maintained during the experimental period. The images were acquired thrice after ensuring that the color map and propagation imaging of the shear wave speed were stable for a few seconds during the session. In the case of pain associated with full extension of the elbows, the elbow joint was slowly and fully extended while consulting the participant to avoid the stretch reflex [2]. Scanning was performed carefully to avoid the pressing or deformation of muscles. The mean shear modulus was calculated over the largest region of interest, excluding the aponeurosis and subcutaneous adipose tissues from the B-mode images. The elastographic images were stored as bitmap (.bmp) files, measured, and averaged using software (iElastographic image analyzer ver. 1, Takei Scientific Instruments Co., Ltd., Kamo, Japan); these data were used for further analyses. All measurements and analyses of the ultrasonographic data were performed by an expert with >10 years of experience. The shear modulus in the resting muscle condition had a coefficient of variation of 2.0% ± 1.9% and an intraclass correlation coefficient of 0.965 (*p* < 0.001).

### 2.9. Titin N-Terminal Fragment Excretion Assay

Approximately 3 mL of urine was collected from each participant, and concentrations of UTF were measured using an enzyme-linked immunosorbent assay kit (Titin *N*-terminal Fragment Assay Kit; Immuno-Biological Laboratories Co., Ltd., Fujioka, Japan) as previously described [21]. Samples were stored at −20 °C. Thawed urine samples were diluted from 1:5 to 1:500 to ensure they were within the linear detection range. Diluted samples and standard solutions were added to each antibody-coated well of 96-well microplates, followed by incubation for 60 min at 37 °C. Subsequently, the microplates were washed four times with a wash buffer. Labeled antibodies were added to each well, and the samples were incubated again for 30 min at 37 °C. After washing five times using wash buffer, microplates were incubated with a tetramethylbenzidine solution for 30 min at room temperature (20–25 °C). Finally, the stop solution was added to each well. The absorbance of the samples was measured at 450 nm using the Multiskan FC microplate reader (Thermo Fischer Scientific, Waltham, MA, USA). The UTF concentration was calculated using a linear regression model, and urinary creatinine levels were estimated using an automated analyzer (Bio Majesty JCA-BM8060; JEOL, Tokyo, Japan). The UTF values were normalized based on urinary creatine (Cr) levels (each raw data point in urine/Cr concentration) [21]. The normal range of UTF concentrations in the general population was considered: 1.47–7.14 pmol/mg/dL [17].

### 2.10. Statistical Analysis

The Shapiro–Wilk test was performed to confirm that baseline values for all items showed normality. Equal variance was assessed using Levene’s test. Given that all variables showed normality and equal variances, the baseline values of each variable were compared between groups using independent *t*-tests. Post-exercise changes in parameters were compared between the groups (PLA vs. CRE) using a two-way ANOVA considering two factors (group × time). In the case of a significant interaction effect, Bonferroni’s post hoc test was performed to identify the time points reflecting significant differences between conditions. The UTF data indicated a non-normal distribution. Therefore, we applied a logarithmic transformation (log_10_) before analysis [19]. All statistical analyses were performed using Predictive Analytics Software version 28 for Windows (SPSS Japan Inc., Tokyo, Japan). Statistical significance was considered at *p* ≤ 0.05.

## 3. Results

### EIMD Indices

The comparative data of physical features recorded in the CRE and PLA groups are depicted in Figure 3. The ROM of the CRE group was higher than that of the PLA group at 24 h post exercise (Figure 3A). MVC was higher in the CRE group than in the PLA group at 0, 48, 96, and 168 h post exercise (Figure 3B). Reduced CIR was recorded in the CRE group compared with the PLA group at 48, 72, 96, and 168 h post exercise (Figure 3C). The SM in the CRE group was lower than in the PLA group at 96 and 168 h post exercise (Figure 3D). No significant differences in UTF or subjective SOR were observed (Figure 3E,G). The CRE group experienced less muscle fatigue compared with the PLA group immediately after exercise and at 168 h post exercise (Figure 3F).

## 4. Discussion

In this study, we investigated whether pre-intake of CrM facilitates recovery from EIMD induced by eccentric exercise. The participants were randomly stratified into the CRE or PLA group based on variables that potentially influence the onset of EIMD, such as age and muscle strength. ROM, MVC, CIR, SM, and muscle fatigue returned to baseline levels more rapidly in the CRE group than in the PLA group.

Both groups exhibited the lowest ROM and MVC values immediately after eccentric exercise (Figure 3A,B) and the highest SOR value at 72 h post exercise (Figure 3G). Yamaguchi et al. [22] reported that MVC decreased by a maximum of 37.3%, ROM declined by 13.6%, and SOR increased by 51.1 mm after isokinetic eccentric exercise performed at 100% effort. In contrast, within the current study, a 73.1% decrease in MVC, a 34.8% reduction in ROM, and a 66.4 mm increase in SOR were detected in the PLA group after isotonic eccentric exercise. However, MVC decreased by 78.7%, ROM declined by 42.1%, and SOR increased by 61.7 mm in the CRE group. A comparison with previously published data suggests that stronger muscle damage was induced in the current study.

We detected significantly higher isometric peak torque at the elbow joint in the CRE group at 0, 48, 96, and 168 h post exercise than in the PLA group (Figure 3B). A meta-analysis of the recovery-facilitating effects of CrM pre-intake reported a modest effect size of 0.81 for muscle force recovery after EIMD [23]. This might have been influenced by the diversity in muscle force measurement techniques (e.g., comparison between vertical jump height and isometric contraction) and the variable targeted muscle groups (e.g., knee extensors versus elbow flexors) considered in different studies. Conversely, a previous study that evaluated both isometric and isokinetic peak torques to assess the CrM-induced recovery [24] reported significant differences in isometric strength but not in isokinetic strength. Similar isometric strength measurements used in the previous and present studies suggest that congruence in contraction type may contribute to the consistency in the results. Additionally, CrM intake facilitates the proliferation and differentiation of satellite cells [25]. Our results indicated that satellite cells show increased activity 24–48 h after high-intensity resistance training. Moreover, a trend toward facilitated recovery at 48 h post exercise was confirmed. Considering this congruence in the recovery time course, we hypothesized that satellite cell activation may facilitate muscle repair, inducing increased mechanical contractile forces and expedited recovery to baseline physical parameters. The present study found a higher degree of EIMD than in previous studies. Therefore, elevated levels of EIMD may further stimulate the activation of satellite cells.

The upper arm circumference was significantly smaller in the CRE group than in the PLA group at 48, 72, 96, and 168 h post exercise (Figure 3C). Damage and inflammatory responses in EIMD can be categorized into primary and secondary events. Primary events involve mechanical destruction of the sarcomeres, followed by secondary events, such as inflammatory and oxidative stress responses [26,27]. These cascading events can lead to increased vascular permeability, intramuscular edema, and leukocyte infiltration, potentially exacerbating damage and inhibiting the structural and functional recovery of muscles [26,27]. Previous studies suggested that CrM exhibits antioxidant [28,29] and anti-inflammatory effects [30,31] that attenuate secondary events following eccentric exercise. Previous in vitro research on rats with pre-administered CrM has confirmed its explicit anti-inflammatory effect on endothelial cells [32]. However, caution must be exercised when directly applying these animal-based findings to humans. We theorize that the dual anti-inflammatory and antioxidant effects of CrM may have contributed to the mitigation of secondary muscle damage responses and the associated reduction in edema in this study.

Significantly lower subjective fatigue levels were demonstrated at 0 and 168 h post exercise during elbow flexion in the CRE group than in the PLA group (Figure 3F). The changes recorded over time revealed lower values in the primary event, unlike that in the circumference. Multifaceted factors contribute to fatigue; however, persistent arm fatigue after exercise is particularly attributable to low-frequency fatigue [2]. This refers to muscle exhaustion induced by sustained exercise or high-intensity short-duration activities; it involves fluctuations in intramuscular calcium ion concentrations [33]. Eccentric exercise impairs excitation–contraction coupling in muscles, leading to an increased concentration of intramuscular calcium ions and low-frequency fatigue [34]. In contrast, CrM enhances the concentration of creatine phosphate in muscle cells and facilitates the rephosphorylation of adenosine triphosphate [35]. This aids in maintaining intracellular calcium homeostasis and the normal function of the sarcoplasmic reticulum calcium pump [36]. Therefore, the results of this study suggest that CrM intake normalizes intramuscular calcium dynamics induced by eccentric exercise, which possibly contributes to the alleviation of subjective fatigue.

No significant differences In UTF were observed between the two groups (Figure 3E). Previous reports on CrM-mediated recovery from EIMD have indicated that the increase in biomarkers that reflect the level of EIMD, such as CK, is attenuated by CrM treatment [37]. In this study, UTF was analyzed as a biomarker. UTF is generated from the fragmentation of the *N*-terminal portion of titin, a major structural protein of myofibrils, owing to mechanical and metabolic damage associated with excessive eccentric contractions, resulting in its leakage into urine [21]. Our study revealed a strong correlation (r = 0.966) between UTF concentration and blood CK activity after EIMD onset [17]. However, in contrast to the response observed using CK, UTF did not significantly differ between the CRE and PLA groups in this study. Yokota et al. [38] evaluated the inhibitory effects of CrM intake on EIMD induced by downhill exercise in mice and detected suppressed inflammation in the test group. However, the increase in UTF was not inhibited, which is comparable to our results. Considering that the mechanisms underlying the release of CK into the bloodstream and the leakage of UTF into the urine differ fundamentally, it can be speculated that CrM may accelerate the recovery from muscle cell membrane damage without significantly affecting titin breakdown.

The CRE group exhibited significantly lower SM values at 96 and 168 h post exercise (Figure 3D). SM reflects passive tension in muscles and increases with an increase in the dorsiflexion angle of the ankle [39]. According to the three-element muscle model proposed by Hill et al. [40], muscle tension involves contractile elements (e.g., actin, myosin, and titin), serial elastic elements (e.g., tendons), and parallel elastic elements (e.g., extracellular matrix). In contrast to the UTF analysis results, the SM differed considerably between groups. As UTF serves as an indicator of contractile elements, the effects of CrM are possibly limited to contractile components and may predominantly influence parallel elastic elements, such as the extracellular matrix. Previous reports suggest that the symptoms of EIMD may be exacerbated when the muscle fascia, a parallel elastic element, undergoes significant elongation during eccentric exercise [41]. Furthermore, Inami et al. [2] pointed out that approximately 50% of alterations in the SM can be explained by UTF, while the remaining 50% is potentially related to the extracellular matrix. Hence, we consider that the accelerated recovery in the SM value of the CRE group may reflect a specific effect of CrM on the extracellular matrix.

Previous meta-analyses assessed the ergogenic effects of dietary supplements in facilitating the prevention of and recovery from EIMD [42]. Markers of inflammation and oxidative stress were significantly reduced within 24–48 h following eccentric exercise in groups that ingested fruit-derived supplements compared with placebo groups [41]. Furthermore, root-vegetable-derived supplements reduce markers of inflammation within 24–48 h post exercise [43]. In contrast, CrM intake exhibited stronger preventive effects against acute symptoms, inflammation, and oxidative stress, particularly within the first 24 h post exercise [43]. The differences observed in these meta-analyses might be attributable to the difference between the biochemical composition of CrM and that of plant-derived extracts. Variables, such as the supplement dosage and the level of muscular strain, can influence outcomes. Hence, it is difficult to confirm that CrM is superior to other options. However, the anti-inflammatory, anti-oxidative stress, and calcium-ion-homeostasis-maintaining effects of CrM indicate that it holds promise as a valuable supplement for managing EIMD.

In the present study, the participant population was limited to young males; hence, potential demographic bias could not be eliminated. Previous research indicates that the degree of EIMD can differ between men and women due to the influence of sex hormones [44]. Therefore, future studies are needed to confirm the effects of CrM using a sample population that includes both sexes. To elucidate the mechanisms underlying the preventative effects of CrM on the signs and symptoms of EIMD, this study employed various metrics, including body composition, muscle stiffness changes, functional muscle assessments, and biochemical markers indicative of muscle cell damage. Nevertheless, future research could benefit from incorporating biomarkers directly involved in muscle damage and repair mechanisms, such as matrix metalloproteinases or Bcl-2-associated athanogene 3. Information on the influence of CrM on these biomarkers can provide a more in-depth understanding of the role of CrM in facilitating recovery from EIMD. These insights will contribute to a comprehensive knowledge of the ergogenic role of CrM in EIMD.

## 5. Conclusions

In this study, we demonstrated that pre-intake of CrM significantly facilitated recovery from EIMD. This improvement was particularly evident in the physiological metrics assessed, including ROM, MVC, arm circumference, SM, and indicators of muscle fatigue. These findings have implications for athletes engaged in consecutive competitions, suggesting that CrM supplementation may be beneficial for accelerating recovery from competition-induced fatigue and muscle damage. The results of this study support the supposition of a previous review article [45] that described the potential of CrM to accelerate recovery, potentially providing new insights for a consensus on the impact of CrM on EIMD.

## Figures and Tables

**Figure 1 nutrients-16-00896-f001:**
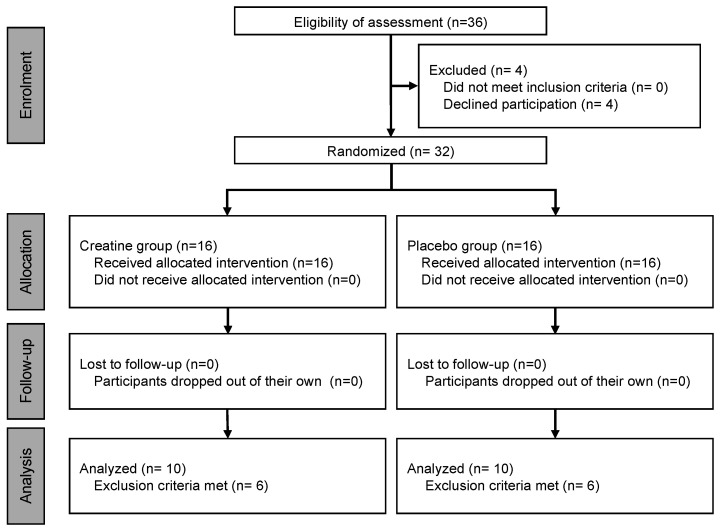
Flowchart showing the parallel randomized controlled trial process involving two groups.

**Figure 2 nutrients-16-00896-f002:**
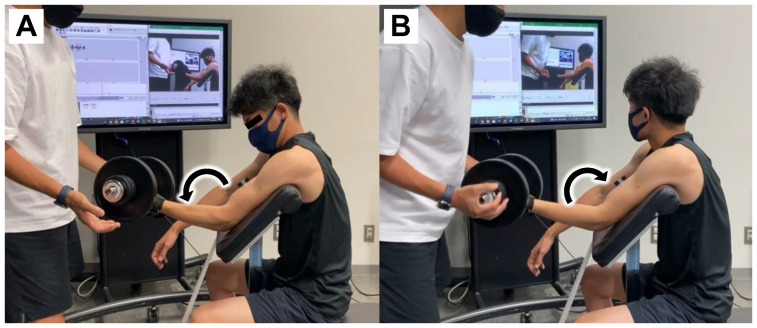
Eccentric exercises. During the down (eccentric) phase (**A**), the participant extends their arms in a controlled manner; the examiner lifts the dumbbells during the up (concentric) phase (**B**).

**Figure 3 nutrients-16-00896-f003:**
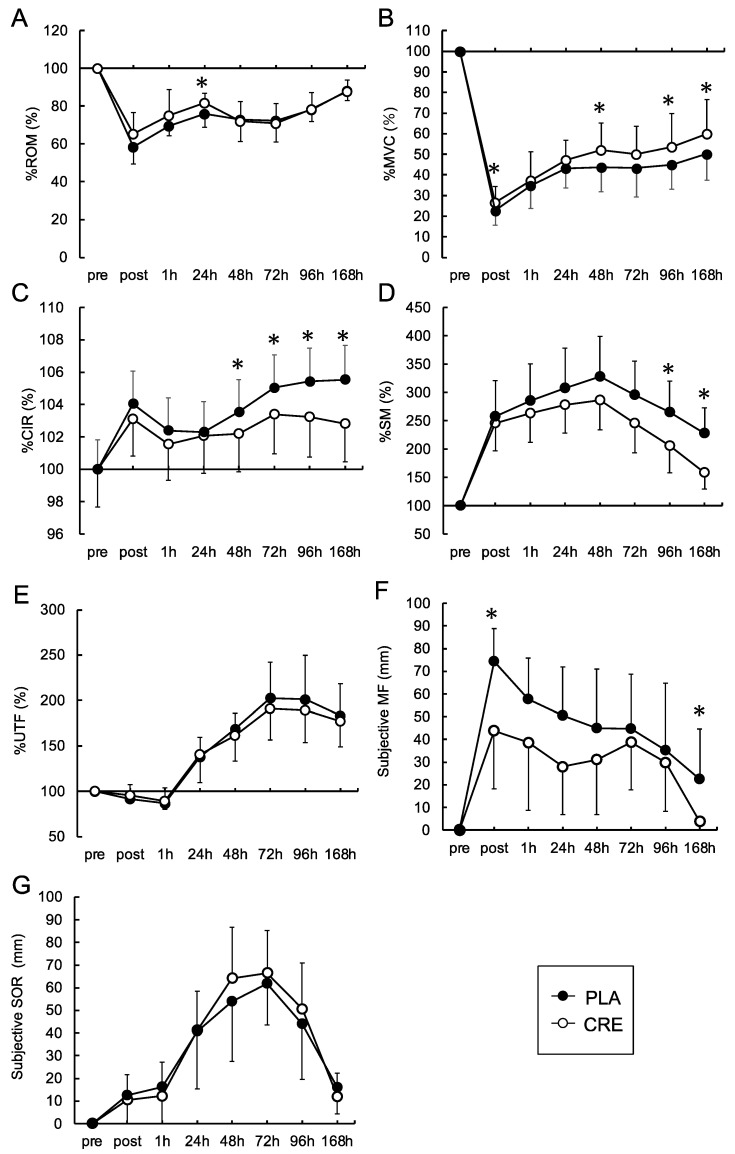
Changes in exercise-induced muscle damage indices over time. PLA, placebo group; CRE, creatine group; ROM, range of motion (**A**); MVC, maximum voluntary contraction (**B**); CIR, circumference (**C**); SM, shear modulus (**D**); UTF, urinary titanium *N*-terminal fragment (**E**); muscle fatigue (**F**); SOR, subjective soreness (**G**). * *p* < 0.05. Data are presented as mean ± SD.

**Table 1 nutrients-16-00896-t001:** Baseline physical characteristics and exercise-induced muscle damage indices.

			Levene’s Test		Independent *t*-Test		
	PLA	CRE	F-Value	*p*-Value	t-Value	*p*-Value	*t*-Test
Age (years)	23.9 ± 1.9	23.5 ± 4.1	1.390	0.252	0.753	0.460	n.s.
Body Mass (kg)	72.1 ± 11.2	70.4 ± 9.9	0.276	0.605	0.293	0.773	n.s.
Body Fat (%)	19.7 ± 4.2	17.5 ± 4.3	0.000	0.995	1.084	0.291	n.s.
SLM (kg)	54.2 ± 6.9	54.5 ± 6.2	0.003	0.959	−0.084	0.934	n.s.
ROM (deg)	123.0 ± 5.3	120.2 ± 2.9	0.002	0.964	0.188	0.852	n.s.
MVC (kgf)	21.6 ± 2.8	21.6 ± 2.2	0.795	0.383	0.070	0.945	n.s.
CIR (cm)	28.6 ± 2.5	27.6 ± 1.8	0.726	0.404	0.703	0.490	n.s.
SM (kPa)	41.4 ± 6.7	45.6 ± 10.3	0.750	0.397	−0.813	0.426	n.s.
UTF/USG	208.8 ± 161.9	313.1 ± 387.3	1.147	0.297	−0.786	0.441	n.s.
SOR (mm)	0.0 ± 0.0	0.0 ± 0.0	−	−	−	−	n.s.
MF (mm)	0.0 ± 0.0	0.3 ± 0.8	5.833	0.027	−1.176	0.255	n.s.

Data are presented as mean ± SD. PLA: placebo group; CRE: creatine group; SLM: soft lean mass; ROM: range of motion; MVC: maximum voluntary contraction; CIR: circumference; SM, shear modulus; UTF: urinary titin *N*-terminal fragment; USG: urine specific gravity; SOR: soreness; MF, muscle fatigue; n.s., not significant.

## Data Availability

The original contributions presented in the study are included in the article, further inquiries can be directed to the corresponding author.

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
