# Peer review of "The Effect of Prior Creatine Intake for 28 Days on Accelerated Recovery from Exercise-Induced Muscle Damage: A Double-Blind, Randomized, Placebo-Controlled Trial"

_nutrients, 2024, doi:10.3390/nu16060896_

Round 1

Reviewer 1 Report

Comments and Suggestions for Authors

I want to thank you for the opportunity to review this manuscript. In general, this is in advance, an interesting research about creatine and muscle damage. Still, after a careful analysis, I believe that this paper, in my opinion, must be modified. Therefore, I hope my recommendations will help you improve the manuscript.

Title

Clear and well-constructed. Can you consider including upper arm in the title? It will work in the same way in the upper leg?

Abstract

Line 21: Please change h by hours (h).

Line 22: Author explain that the lower upper arm circumference is measured in post exercise situation, was it measured pre-exercise?

Introduction

The authors do not justify eccentric exercise in the introduction chapter.

Lines 40-43 are about results of 15 male college soccer players (mean age, 21.3 ± 2 years, height 172.2 ± 6.6 cm, weight 66.1 ± 11 kg and BMI 22.3 ± 3.2) from Jamia Millia Islamia, New Delhi, India. This cannot support your results.

Materials and methods

The predetermination of the sample size states 167 subjects, but as previous similar studies use 10, it is decided to use 10. Would it be possible to assess the alpha error and statistical power for an n of 10?

Has the sample been matched for age or other variables of interest to the study?

Line 101 authors inform the CRE intake, 3 g. Is it per day? Maybe you can explain better de posology

Line 207 authors inform about the use of Levene test, but they do not include normality tes (Saphiro-Wilks, or similar).

Line 208 you use t-test, I suppose is independent t-test (for comparison of CRE and PLA groups)

In general, the methods chapter is extensive and well structured

Results

In methods you say that authors use “using a two-way ANOVA considering two factors (group × time)”, but there are not table (results) about this question.

In general, the results are correct but do not correspond exactly with the material and methods chapter. I consider that the statistical analysis part should be modified for a better prior understanding of what is being done.

Discussion

Complete and well-structured.

Conclusions

Similar to this paper in “Nutrients” journal:

Wax, B.; Kerksick, C.M.; Jagim, A.R.; Mayo, J.J.; Lyons, B.C.; Kreider, R.B. Creatine for Exercise and Sports Performance, with Recovery Considerations for Healthy Populations. Nutrients 2021,13, 1915.https://doi.org/10.3390/nu13061915A

Kindest Regards

Reviewer 2 Report

Comments and Suggestions for Authors

Abstract

The abstract seems quite basic and the language used is at times not very scientific. For example you say "significantly higher " but there are no levels of statistical significance and nor are there any data presented.

Introduction

The topic is well researched in the literature so I found the Introduction somewhat lacking with respect to Creatine (CrM) ingestion. This could be expanded to cover more fully the area with respect tot he work that you did. Specifically you need to examine more closely how the intake of CrM will improve or not the activity that you undertook.

Lines 66-68 could I think be deleted - they are not part of the Introduction.

Methods

You criticise small sample size in the Introduction but this is not a big sample size, more would have been better I think! 

You need to cite some literature supporting you exclusion criteria!

Line 102 - is inoculate the right word?

A diagram or picture of the exercises used would be useful possibly? 

Maybe the use of an isokinetic dynamometer would be better ?  (I understand that this might not have been available!) 

Try not to start sentences with an abbreviation, with And or For, of From - correct throughout please. 

Results

Baseline data should go in the methods - as they have not been tested yet it cannot be part of the results of the experiment.

I thought there were aspects of the Results that could have been referred to in the abstract!  

I liked Figure 2 but would like to see it larger??  Difficult to read and get any real perspective on what changed. Figures are clear BUT need to be larger. 

Discussion

The 1st and some of the second paragraph could be deleted - simply re-reports what you did in too long a manner.  Needs to be shortened.

I have no other comments regarding the Discussion which I thought for the most part was well written and appropriate. 

Conclusions

Brief and to the point ! 
